# *Bacillus** thuringiensis* Exopolysaccharide BPS-2 Ameliorates Ulcerative Colitis in a Murine Model Through Modulation of Gut Microbiota and Suppression of the NF-κB Cascade

**DOI:** 10.3390/foods14132378

**Published:** 2025-07-04

**Authors:** Zexin Gao, Huan Li, Jungang Wen, Wenping Ding, Jie Yu, Yue Zhang, Xiaojuan Song, Jianrong Wu

**Affiliations:** 1Engineering Research Center of Medical Biotechnology, School of Biology and Engineering, Guizhou Medical University, Guiyang 550025, China; wenjungang@163.com (J.W.); dingwenping19@mails.ucas.ac.cn (W.D.); yujie@gmc.edu.cn (J.Y.); zhangyue0816@gmc.edu.cn (Y.Z.); 2Key Laboratory of Carbohydrate Chemistry, School of Biotechnology, Biotechnology of Ministry of Education, Jiangnan University, Wuxi 214122, China; lh892023@163.com; 3The Key Laboratory of Environmental Pollution Monitoring and Disease Control of Ministry of Education, School of Public Health, Guizhou Province Engineering Research Center of Health Food Innovative Manufacturing, Guizhou Medical University, Guiyang 550025, China; songxj1029@163.com

**Keywords:** *Bacillus thuringiensis*, gut microbiota, ulcerative colitis, molecular docking, food additive

## Abstract

This study investigated the therapeutic potential of *Bacillus thuringiensis* extracellular polysaccharide BPS-2 in dextran sulfate sodium (DSS)-induced ulcerative colitis (UC) murine models. BPS-2 demonstrated significant efficacy in ameliorating UC-associated pathologies through three principal mechanisms: (1) attenuating histopathological damage while preserving colon epithelial integrity, (2) modulating immune marker expression patterns in colon tissues, and (3) restoring gut microbiota homeostasis. BPS-2 exhibited multi-faceted protective effects on the gut by mitigating oxidative stress responses and enhancing short-chain fatty acid biosynthesis, leading to an improved gut microbial community structure. Molecular docking analysis displayed strong binding affinity (ΔG = −7.8 kcal/mol) between the BPS-2U fragment and the Nuclear Factor κB (NF-κB) p50/p65 heterodimer, suggesting the potential disruption of NF-κB signaling pathways. Complementary molecular dynamics simulations revealed exceptional conformational stability in the p65-BPS-2U complex. These findings establish BPS-2 as a natural food additive that modulates the microbiota-barrier–inflammation axis through dietary intervention, offering a novel strategy to alleviate UC.

## 1. Introduction

UC, a chronic subtype of inflammatory bowel disease (IBD), features immune dysregulation and recurrent colonic mucosal ulcers, clinically manifesting as diarrhea, hematochezia, and weight loss [1,2]. Globally, UC incidences have demonstrated a progressive upward trajectory with a particularly pronounced increase in industrialized nations, where prevalence rates now exceed 0.5% [3]. Conventional pharmacological interventions, including aminosalicylates, corticosteroids, and immunosuppressants, have demonstrated moderate efficacy in managing symptoms [4]. However, long-term administration increases the risk of infections and causes metabolic disturbances and neuropsychiatric manifestations [5]. This underpins the need for prompt exploration of functional dietary components and symptom-alleviating strategies as adjunctive/alternative therapies [6].

Natural products have drawn considerable attention in pharmaceutical research due to their multi-target modulating capacity and the advantage of low toxicity. Among these bioactive compounds, polysaccharides have emerged as promising candidates for UC therapy, demonstrating remarkable anti-inflammatory properties, immunomodulatory effects, and intestinal barrier restoration capabilities [3]. These complex carbohydrates exhibit diverse biological activities, which are attributed to their structural heterogeneity [7]. Mechanistic studies reveal that Astragalus and *Ganoderma lucidum* polysaccharides exert therapeutic effects by modulating Nuclear Factor κB (NF-κB) signaling pathways, effectively suppressing pro-inflammatory cytokines, including tumor necrosis factor alpha (TNF-α) and interleukin-6 (IL-6) [8,9]. Marine-derived polysaccharides, particularly alginates, demonstrate distinct mechanisms of action, particularly through the upregulation of intestinal epithelial tight junction (TJ) proteins (zonula occludens-1 (ZO-1) and Occludin) to enhance mucosal barrier integrity [10]. Furthermore, dietary fibers (pectin/β-glucan) ameliorate UC via microbiota-mediated short-chain fatty acids (SCFAs), restoring gut homeostasis and mitigating disease progression [11].

In recent years, microbial metabolites, particularly exopolysaccharides (EPS), have emerged as a research focus on identifying new UC therapeutics due to their anti-inflammatory, immunomodulatory, and gut barrier-restoring properties [12]. Studies have demonstrated that *Lactobacillus*-derived EPS ameliorate colon tissue damage through NF-κB signaling pathway modulation, effectively suppressing pro-inflammatory cytokines [13]. Furthermore, *Bacteroides*-derived polysaccharides enhance mucosal integrity and restore gut microbiota homeostasis by promoting commensal bacterial proliferation and modulating SCFAs metabolism [14]. Mechanism-based investigations showed that this bioactive material significantly reduces NF-κB phosphorylation, which downregulates downstream pro-inflammatory cytokines (TNF-α, interleukin-1 beta (IL-1β), and IL-6) and improves oxidative stress-mediated inflammatory cascades [15]. EPS demonstrates multi-target therapeutic potential in UC management; however, challenges like unclear structure–activity relationships, suboptimal delivery systems, and clinical translation gaps still persist.

This study focused on the UC attenuation activity of the EPS BPS-2 from *Bacillus thuringiensis* IX-01. BPS-2 was previously reported as a glycosaminoglycan (GAG) derivative, and its primary chemical structure has been resolved [16]. In vitro studies confirmed the anti-inflammatory activity of BPS-2. As GAGs play an important role in tissue expansion, intestinal stabilization, and barrier maintenance [3], we propose that BPS-2 preserves intestinal barrier homeostasis and alleviates UC-related gut microbiota dysbiosis (Figure 1). Preliminary evidence from in vitro dynamic gastrointestinal reactor simulations demonstrated the regulatory effects of BPS-2 on the gut microbiota composition of UC patients [17]. However, in vitro studies on BPS-2 efficacy present certain limitations that necessitate in vivo validation. This study assessed the effects of BPS-2 on intestinal mechanical/immune barrier restoration in a DSS-induced UC mouse model. BPS-2 driven mechanisms were elucidated using integrated analyses of gut microbiota, cytokine profiles, and barrier-related protein expression, complemented with computational simulations of its structural domains. This study elucidated the BPS-2 structure–activity relationship in modulating diet-mediated intestinal inflammation concomitantly, establishing theoretical and practical frameworks to develop BPS-2 as a functional food ingredient with mucosal barrier restoration properties.

## 2. Materials and Methods

### 2.1. Reagents and Preparation of BPS-2

Biochemical analysis kits, including those for myeloperoxidase (MPO), malondialdehyde (MDA), catalase (CAT), glutathione peroxidase (GSH-Px), superoxide dismutase (SOD), inducible nitric oxide synthase (iNOS), cyclooxygenase-2 (COX-2), and fecal occult blood detection were sourced from Nanjing Jiancheng Bioengineering Institute (Nanjing, China). Tight junction protein detection kits specifically measuring ZO-1, Occludin, and claudin-1 levels were acquired from Nanjing Senbeiga Biotechnology Co., Ltd. (Nanjing, China). DSS was purchased from MP Biomedicals LLC (Santa Ana, CA, USA). The enzyme-linked immunosorbent assay (ELISA) kits for interleukin-10 (IL-10), IL-6, TNF-α, and IL-1β were purchased from Jiangsu Meimian Biotechnology Co., Ltd. (Changzhou, China).

*Bacillus thuringiensis* IX-01 (source of BPS-2) was isolated from Sichuan’s Pixian bean paste, and the strain was deposited at the China Center for Type Culture Collection (M 2020486). The 16S rRNA sequence was deposited into GenBank at NCBI with the accession number OL687443. The preparation of BPS-2 was based on the method described by Gao et al. [16] and is summarized in Appendix A. Gao et al. [16] also successfully determined the structure of BPS-2 (Appendix A).

### 2.2. Animals

Thirty male C57BL/6J mice (8 weeks old, SPF grade) were purchased from Beijing Vital River and housed in a controlled Specific-Pathogen-Free (SPF) facility (23 ± 3 °C, 40%–70% humidity, 12 h light/dark cycle) at Jiangnan University. Standard chow was provided by the institution’s Laboratory Animal Center. All animal-related procedures complied with the European Union Directive 2010/63/EU and were approved by Jiangnan University’s Institutional Animal Care and Use Committee (JN. No20220915c0401030[316]).

### 2.3. Establishment of UC Mice Models and BPS-2 Treatment

In accordance with the experimental design (Figure 2A), C57BL/6J mice were randomly divided into five groups (*n* = 6), which were the control group, the DSS group, the low-dose BPS-2 group (BPS-2-L), the medium-dose BPS-2 group (BPS-2-M), and the high-dose BPS-2 group (BPS-2-H). The mice were initially allowed to acclimatize to the conditions within the animal facility for 7 days. The subsequent 14-day period was designated as the experimental phase, with each group of mice receiving different treatments as described below:

In this 14-day study, five experimental groups were established: (I) Control mice received a normal diet with daily 200 μL saline gavage; (II) the DSS group received 2.5% DSS solution in their drinking water on days 8–14 via saline gavage; (III–V) BPS-2 treatment groups (L/M/H) that were subjected to the DSS protocol received 200 μL BPS-2 solution at 50, 100, or 200 mg/kg daily doses, respectively, via gavage on days 8–14 [18]. All groups had ad libitum access to a standard diet throughout the experimental period, with interventions specifically timed to evaluate DSS-induced effects and BPS-2 dose responses.

### 2.4. Animal Dissection

On day 14, mice were anesthetized with 3% isoflurane, and blood was drawn from the orbital sinus. Blood samples were left standing for 60 min to facilitate natural coagulation, then centrifuged, and serum was separated. Sera samples were stored at −80 °C until required. Once blood had been collected, mice were euthanized by cervical dislocation. The abdominal colon was subsequently excised and subjected to the histopathological evaluation of tissue sections to assess the severity of colonic injury, with disease activity scores systematically assigned based on standardized criteria [19].

### 2.5. Evaluation of Disease Activity Index (DAI)

The DAI score, calculated as the cumulative sum of weight loss, fecal consistency, and fecal occult blood scores, was adapted from Hu et al. [20] to assess colitis severity. During the establishment of the DSS-induced colitis mice models, animals were weighed prior to intragastric administration at a consistent time each day. Fecal samples were collected via sterile metabolic cages and promptly flash-frozen. The detection of fecal occult blood was carried out in accordance with the manufacturer’s instructions. A detailed breakdown of the scoring coefficients can be found in Appendix A.

### 2.6. Histopathological Analysis and Injury Scoring of Colonic Tissue

The histopathological evaluation of colitis severity primarily relies on tissue section analysis and standardized scoring systems. Distal colon specimens from mice were fixed in 4% paraformaldehyde solution for 48 h before undergoing standard histological processing, including dehydration, paraffin embedding, and sectioning. For the objective assessment of colonic damage, haematoxylin and eosin (H&E)-stained sections were evaluated according to the histopathological scoring system established by Zhang et al. [21], which quantitatively assesses lesion extent, crypt destruction, lesion depth, and inflammatory severity (specific criteria are provided in Appendix A).

### 2.7. Biochemical Assays

The supernatant of mouse colonic tissue was collected, and protein concentrations were determined with the bicinchoninic acid protein assay kit. The levels of COX-2, ZO-1, IL-10, Claudin-1, IL-6, iNOS, IL-1β, TNF-α, and Occludin in the supernatant of mouse colon tissue were detected using commercial ELISA kits. The determination of myeloperoxidase, malondialdehyde, superoxide dismutase, catalase, and glutathione peroxidase was undertaken based on the manufacturer’s (cf) instructions.

### 2.8. Immunohistochemical Analysis

Immunohistochemical staining was performed according to Yang et al. [22]. Briefly, paraffin-embedded colon sections were deparaffinized, rehydrated, and washed with PBS. Antigens were retrieved using a microwave while 3% H_2_O_2_ blocked endogenous peroxidases. The sections were then blocked in 3% Bovine Serum Albumin and incubated with primary (4 °C/overnight) and HRP-conjugated secondary antibodies (room temperature/50 min). DAB chromogen with hematoxylin counterstaining was used to visualize the target antigens. Stain intensity was quantified with ImageJ 1.53e (National Institutes of Health, Bethesda, MD, USA) software, and the integrated optical density was normalized to tissue area (IOD/mm^2^).

### 2.9. SCFAs Analysis and 16S rRNA Gene Sequencing

SCFA quantification was performed using external standard calibration curves established with SCFAs standards using an Agilent 7890A gas chromatograph (Agilent Technologies, Santa Clara, CA, USA) equipped with an HP-INNOWax capillary column [23]. Fecal DNA was extracted using the QIAamp PowerFecal Pro DNA Kit (Qiagen, Germantown, MD, USA) followed by the amplification of the 16S rRNA V3-V4 region with 338F/806R primers. Sequencing data were analyzed according to Gao et al. [24].

### 2.10. Structural Fragments Construction and Molecular Docking

The structural fragment of the decomposed oligosaccharide (BPS-2) was initially constructed using the CHARMM-GUI online platform (version 1.8, accessible at https://charmm-gui.org (accessed on 15 March 2025)) [25] following a standard protocol for carbohydrate modeling. The three-dimensional coordinates of the NF-κB-DNA complex (PDB ID: 1LE5) were retrieved from the RCSB Protein Data Bank. Molecular visualization and preliminary structure building were performed using PyMOL 2.6 (Schrödinger LLC, New York, NY, USA), including geometric optimization through the following sequential steps: (1) the removal of water molecules, (2) the elimination of redundant heteroatoms and DNA, and (3) the energy minimization of the protein. Subsequently, molecular docking simulations were conducted using AutoDockTools 1.5.7 software suite. Subsequent molecular interaction analyses were performed using PyMOL and Discovery Studio 2019 Visualizer (Dassault Systèmes, Shanghai, China).

### 2.11. Molecular Dynamics Simulation Computation

The molecular dynamics (MD) simulations were performed using the Gromacs 2022 software package with a 100 ns production run. The CHARMM36 force field was employed for protein parameterization, while ligand topology was generated using the General Amber Force Field 2 (GAFF2) [25]. The protein–ligand complex was solvated in a TIP3P water-filled cubic periodic box (1.2 nm buffer). To determine van der Waals and Coulombic interactions, particle mesh Ewald (PME, 0.12 nm Fourier spacing) was used for long-range electrostatics with a Verlet-cutoff truncation at 1.0 nm. Sequential equilibration involved NVT (50,000 steps, 100 ps, τ = 0.1 ps) followed by NPT (50,000 steps) using the Berendsen thermostat–barostat (310 K, 1 bar). Production simulations (100 ns, NPT) utilized the Parrinello–Rahman barostat with a 2 fs timestep, LINCS bond constraints, and trajectory sampling every 10 ps.

### 2.12. Statistical Analyses

Statistical comparisons between groups were performed using one-way analysis of variance (one-way ANOVA) followed by Duncan’s multiple range test for post hoc pairwise comparisons. Analyses were conducted using SPSS Statistics 26.0 software (IBM Corp., Armonk, NY, USA). Experimental results are presented as mean ± standard deviation (mean ± SD). Statistical significance was defined as follows: *p* < 0.05, * *p* < 0.01, ** *p* < 0.001, *** *p* < 0.0001. In the analysis of colonic microbial diversity in mice, we first calculated α-diversity indices using QIIME software (v1.9.1). The statistical analysis of intergroup differences in α-diversity was performed using R software (v2.15.3), with Tukey’s test applied for multiple comparisons. Subsequently, β-diversity was assessed by computing Unifrac distances using QIIME (v1.9.1), followed by intergroup difference analysis in R (v2.15.3). Principal component analysis (PCA) was employed to visualize differences in microbial community structure, with the PCA computation implemented using the ade4 package and graphical visualization performed using the ggplot2 package in R (v2.15.3). Intergroup significance testing for β-diversity was similarly conducted using Tukey’s test.

## 3. Results and Discussion

### 3.1. BPS-2 Improves Pathological Indicators in UC Mice

An experimental colitis mouse model was established by induction with DSS. In the established model, typical pathological features were observed on the 7th day of the experiment when compared to the control: a significant decrease in body weight (16.55%), shortening of the colon length, and bloody stools (Figure 2B). Based on this, the effects of BPS-2 polysaccharide intervention were systematically monitored for body weight changes, DAI, and colon morphology to evaluate its potential efficacy in alleviating colitis. After 14 days of BPS-2 polysaccharide intervention, all dose groups exhibited protective effects to varying degrees. Notably, the BPS-2-H group (200 mg/kg/d) demonstrated the best intervention effect, significantly reducing the body weight loss to 8.45% (*p* < 0.01). The BPS-2-L and BPS-2-M groups (50 and 100 mg/kg/d, respectively) also showed a dose-dependent protective trend, with body weight loss rates controlled at 12.3% and 9.8%, respectively. This finding is in good agreement with previous reports. For example, EPS produced by *Bacillus subtilis* significantly alleviated body weight loss in DSS-induced colitis mice by regulating the gut microbiota [22]. In addition, compared with the DSS group, the galacto-oligosaccharide intervention group derived from *Bacillus circulans* reduced the body weight loss rate by 35.7%, further supporting the potential value of microbial-derived polysaccharides in colitis intervention [19].

DAI serves as a critical biomarker to evaluate both the success of DSS-induced colitis modeling and the therapeutic efficacy of bioactive compounds. As shown in Figure 2C, DSS-treated mice exhibited severe colitis phenotypes on day 7, with a mean DAI score of 11, characterized by pronounced hematochezia (rectal bleeding), 16.55% body weight loss, and liquid stool consistency. These pathological manifestations align with previously reported DSS-induced colitis models showing 19.14% weight loss and hemorrhagic diarrhea [22], confirming the successful establishment of colitis in this study. BPS-2 administration significantly attenuated disease progression in a dose-dependent manner. Compared to the DSS group, the BPS-2-L (50 mg BPS-2/kg/day), BPS-2-M (100 mg/kg/day), and BPS-2-H (200 mg/kg/day) groups exhibited a reduction in DAI scores of 30.27%, 33.36%, and 34.68%, respectively (*** *p* < 0.001 vs. DSS group). Notably, while all treatment groups demonstrated comparable therapeutic effects with no statistically significant intergroup differences (*p* > 0.05), the BPS-2-H group (200 mg/kg/day) exhibited the most pronounced ameliorative effect. BPS-2 exhibits dose-dependent amelioration of colitis-associated pathologies, most likely through anti-inflammatory action and mucosal barrier reinforcement. This is supported by robust DAI suppression across the treatment cohorts, indicating therapeutic potential for UC remission.

The length of the mouse colon is strongly correlated with the intensity of inflammatory responses and serves as a critical indicator for assessing colitis severity [12]. Following euthanasia, complete colon tissues were excised and their lengths measured (Figure 2E). As shown in Figure 2D, DSS-treated mice demonstrated a mean colon length of 3.85 cm, which represented a statistically significant reduction (*p* < 0.0001) compared to the 6.39 cm average length observed in the control group. The reported normal colon length of healthy C57BL/6J mice is approximately 7 cm [18], confirming the successful establishment of the colitis model using 2.5% DSS in C57BL/6J mice. Notably, the administration of BPS-2 resulted in dose-dependent therapeutic effects on colon shortening. The BPS-2-H group (200 mg/kg/day BPS-2) showed optimal efficacy, with treated animals maintaining an average colon length of 5.23 cm (Figure 2D). These findings collectively demonstrate that BPS-2 polysaccharide derived from *B. thuringiensis* IX-01 significantly ameliorates pathological indicators of DSS-induced colitis in C57BL/6J mice.

### 3.2. BPS-2 Alleviates Histopathological Damage in UC Mice

As shown in Figure 3A, the histopathological evaluation of mouse colon tissues through H&E staining revealed distinct morphological differences between experimental groups. The control group exhibited intact colonic mucosa with well-aligned epithelial cells and well-preserved crypt architecture. Notably, these specimens demonstrated prominent goblet cell enrichment in the mucosal layer without evidence of inflammatory cell infiltration or erosive changes. In contrast, the DSS-induced model mice displayed significant mucosal damage characterized by submucosal edema, partial epithelial disintegration, and crypt loss (Figure 3A). Additional pathological manifestations included mucosal layer erosion and the substantial depletion of goblet cells. Consistent with previous studies, DSS administration induced characteristic inflammatory changes including crypt structure disappearance, inflammatory cell infiltration, and edema in the serosal layer, confirming the successful establishment of the colitis model [21].

Among all experimental groups, the high-dose BPS-2 regimen demonstrated optimal colon protective effects, preserving crypt architecture integrity without a significant loss of goblet cells (Figure 3A). Histopathological evaluation revealed moderate edema and comparatively higher inflammatory cell infiltration in the submucosal and serosal layers of low- (8.67 ± 2.34) and medium-dose (8.00 ± 2.61) treatment groups. These findings align with previous reports utilizing colon histopathology scoring as a critical evaluation parameter for exopolysaccharide-mediated colitis mitigation [26]. The DSS-induced colitis model group exhibited significantly elevated colon injury scores (12.5 ± 1.87) compared to the control group (2.67 ± 0.82) (*p* < 0.0001) (Figure 3B). Low- (50 mg/kg/day) and medium-dose (100 mg/kg/day) BPS-2 polysaccharide interventions significantly ameliorated colonic inflammation in colitis mice (*p* < 0.01), while high-dose treatment (200 mg/kg/day) showed remarkable efficacy in reducing histopathological damage (*p* < 0.001).

The impact of BPS-2 polysaccharide on colonic inflammatory enzyme activities was evaluated through measurements of MPO, COX-2, and iNOS activities. MPO activity in murine colon tissue serves as a quantitative indicator of neutrophil infiltration, with its protein expression levels directly correlating with the severity of colonic inflammation [27]. The administration of low-, medium-, and high-dose BPS-2 polysaccharide significantly reduced DSS-induced MPO activity from 5.08 ± 0.67 to 2.74 ± 0.59, 2.28 ± 0.31, and 2.87 ± 0.27 U/g protein, respectively (Figure 3C). All dosage groups exhibited a highly significant suppression of MPO protein secretion (*p* < 0.001), consistent with DAI and colon length measurements. While COX-2 activity in all treatment groups showed modest reduction compared to the DSS group, only the high-dose BPS-2 intervention demonstrated statistically significant attenuation (*p* < 0.05) (Figure 3D). Notably, iNOS activity displayed the most pronounced response, with the DSS group showing peak activity at 8.6 ± 1.14 U/g protein. BPS-2 intervention at low, medium, and high doses dramatically suppressed iNOS activity to 5.46 ± 1.14, 3.93 ± 0.76, and 3.18 ± 0.57 U/g protein, respectively (*p* < 0.001 vs. DSS group) (Figure 3E). This finding is biologically relevant, as elevated iNOS expression has been associated with T-cell activation arrest and apoptosis-critical considerations given the fundamental role of T cells in immune system regulation [28].

### 3.3. BPS-2 Maintains the Colonic Epithelial Cell Layer in UC Mice

Colonic mucosal epithelial cells maintain the intestinal mechanical barrier where the structural and functional properties of the barrier are critically dependent on specialized intercellular junctions, particularly tight junctions (TJs) and adherens junctions (AJs). These junctional complexes mediate firm adhesion between intestinal epithelial cells and the intestinal barrier, collectively regulating the paracellular movement of solutes, fluids, and luminal microorganisms while preventing pathogenic microbial invasion [29]. To determine the impact of BPS-2 polysaccharides on TJ proteins in the colonic epithelial layer, we employed immunohistochemical analysis and quantitative ELISA to measure expression levels of the key TJ markers ZO-1, Occludin, and Claudin-1 in colon tissues (Figure 4).

As illustrated in Figure 4A–C, the DSS-induced colitis model group exhibited significant reductions in ZO-1, Occludin, and Claudin-1 protein expression levels, accompanied by the near-complete disruption of intestinal epithelial junctional complexes. This impairment of intestinal mucosal barrier integrity substantially increased susceptibility to colitis [10]. In contrast to the DSS group, control mice maintained higher expression levels of TJ proteins and preserved colon barrier integrity, as evidenced by more intense immunohistochemical staining intensity across all protein markers. Notably, BPS-2 polysaccharide intervention demonstrated dose-dependent therapeutic effects, with progressively intensified TJ protein immunostaining and improved colonic barrier integrity observed with increasing polysaccharide concentrations. These findings suggest that BPS-2 polysaccharide enhanced ZO-1, Occludin, and Claudin-1 protein expression, thereby facilitating restoration of the colonic epithelial barrier and ameliorating UC pathology [30]. Complementary analysis using ImageJ software revealed quantitative increases in TJ protein optical density (Figure 4D–F), which was positively correlated with TJ protein concentrations as measured by ELISA (Figure 4G–I). These results demonstrate that BPS-2 polysaccharide exerts protective effects against colitis through the regulation of TJ protein expression and the subsequent restoration of the colonic mechanical barrier.

### 3.4. BPS-2 Regulates Colonic Immune Markers in UC Mice

*B. thuringiensis* BPS-2 polysaccharide modulated DSS-colitis inflammatory cytokines in mice, with ELISA profiles revealing elevated IL-1β/IL-6/TNF-α and suppressed IL-10 in the model groups (Figure 5). A previous study demonstrated that 1,6-α-D-glucan modulates TNF-α and IL-1β expression through the suppression of NLPR3 (NOD-, LRR- and pyrin domain-containing protein 3) and NF-κB p65 protein expression, suggesting that this mechanism may contribute to the observed anti-inflammatory effects of BPS-2 in UC model mice [31].

Elevated TNF-α levels have been associated with intestinal barrier dysfunction and mucosal inflammation, while excessive IL-6 production during inflammatory responses can trigger hyperactive immune reactions [32]. Our experimental data revealed a dose-dependent reduction in pro-inflammatory cytokine (IL-1β, IL-6, and TNF-α) production following BPS-2 intervention, indicating its potent inhibitory effects on pro-inflammatory cytokine secretion in the colon. As shown in Figure 5C, BPS-2 administration significantly upregulated IL-10 expression in a concentration-dependent manner (*p* < 0.01). This finding aligns with established evidence that IL-10 facilitates immune cell differentiation, antagonizes pro-inflammatory cytokine production, and suppresses adhesion molecule expression [22]. Our findings demonstrate that BPS-2 intervention effectively ameliorates UC progression through dual mechanisms: the suppression of pro-inflammatory cytokines (IL-1β, IL-6, and TNF-α) and the enhancement of anti-inflammatory IL-10 expression.

### 3.5. BPS-2 Reduces Oxidative Stress in UC Mice

During prolonged UC, the affected organism exhibits hyperactive immune responses accompanied by the excessive production of reactive oxygen species (ROS), leading to the disruption of antioxidant homeostasis [33]. Previously, Gao et al. [16] reported on the immunomodulatory effects of BPS-2 polysaccharides on RAW 264.7 phagocytes in vitro. In the current investigation, we evaluated the impact of BPS-2 on oxidative stress in murine models by measuring the enzymatic activities of CAT, MDA, SOD, and GSH-Px in the colon.

Medium- and high-dose BPS-2 administration significantly elevated CAT expression compared to the DSS-induced model group (*p* < 0.05) (Figure 5E). Catalase serves as a natural antagonist against ROS and mycotoxins by eliminating harmful metabolic byproducts [7]. Hence, catalase levels directly reflect an organism’s capacity to scavenge ROS. Our in vivo findings corroborate previous in vitro observations of BPS-2 neutralizing ROS molecules. MDA is a key lipid peroxidation metabolite whose levels inversely correlate with colon tissue integrity [18]. The DSS-induced UC model exhibited the highest MDA content (1.94 ± 0.18 nmol/g protein), while the medium-dose BPS-2 intervention (100 mg/kg/day) significantly reduced MDA expression (*p* < 0.05), indicating the repair of the original damage observed in the colon (Figure 5F). SOD activity was also enhanced in a dose-dependent manner in the presence of BPS-2 (*p* < 0.01, Figure 5G), with high-dose treatment increasing SOD activity by 1.21-fold to 41.64 ± 2.32 U/mg protein compared to the DSS control. This aligns with established correlations between SOD activity and ROS clearance capacity, implying that BPS-2 negatively modulates ROS production through SOD upregulation [34]. GSH-Px is a ubiquitous peroxidase catalyzing hydrogen peroxide decomposition and glutathione recycling to maintain cellular membrane integrity by detoxifying peroxides into harmless hydroxyl compounds [4]. Figure 5H demonstrates that all BPS-2 doses induced significant increases in GSH-Px activity compared to control models (*p* < 0.001). Collectively, BPS-2 polysaccharides ameliorated oxidative stress-induced colon damage in the UC models through the coordinated upregulation of antioxidant enzymes (SOD, CAT, and GSH-Px) coupled with suppression of lipid peroxidation (MDA).

### 3.6. BPS-2 Promotes the Secretion of SCFAs in UC Mice

Although previous in vitro fermentation experiments have demonstrated that BPS-2 primarily yields acetate, propionate, and butyrate, with only trace amounts of valerate detected, current literature underscores the therapeutic potential of valerate in murine models [17]. Specifically, valerate can alleviate inflammation and autoimmune diseases by promoting the secretion of IL-10 and inhibiting the apoptosis of regulatory B cells [35].

As illustrated in Figure 6A, both the control group and high-dose BPS-2 group exhibited significantly higher acetate concentrations compared to DSS-induced colitis mice (*p* < 0.05). Acetate production showed a dose-dependent trend with increasing BPS-2 concentrations, although increases for the BPS-2-L and BPS-2-M groups were not statistically significant (*p* > 0.05). Acetate, the most abundant SCFA in gut microbiota metabolism, plays a pivotal role in regulating intestinal pH homeostasis, supporting butyrate-producing bacterial proliferation, and enhancing probiotic community diversity [3]. All BPS-2 intervention groups showed significantly elevated propionate levels versus the DSS group (*p* < 0.05, Figure 6B), with concentrations of 44.95 ± 5.48, 54.57 ± 9.65, and 50.63 ± 10.07 mmol/L, respectively. This aligns with reported mechanisms where propionate modulates a balance in T_reg_ cells to ameliorate systemic inflammation in hypertension and atherosclerosis [36]. Medium- and high-dose BPS-2 interventions significantly enhanced butyrate and valerate production in colitis mice compared to DSS controls (*p* < 0.01) (Figure 6C,D) The 200 mg/kg/day BPS-2 treatment resulted in butyrate and valerate levels of 24.04 ± 3.81 mmol/L and 23.33 ± 4.14 mmol/L, respectively. These findings are in line with the data on IL-10 secretion (Figure 5C), demonstrating that higher BPS-2 doses potentiate IL-10 expression through the dual induction of butyrate and valerate production in UC mice. High-dose BPS-2 achieved peak total SCFA production (198.42 ± 11.1 mmol/L), representing a 157.79% increase over DSS model levels (Figure 6E).

### 3.7. BPS-2 Improves the Gut Microbiota of UC Mice

Although the precise pathogenesis of UC remains unclear, accumulating evidence has highlighted the critical role of gut microbiota dysbiosis in UC development [5]. Operational taxonomic units (OTUs) clustering (97% similarity) identified 607 (control), 499 (DSS), 470 (BPS-2-L), 525 (BPS-2-M), and 556 (BPS-2-H) OTUs. BPS-2-H restored microbial richness in UC mice, while BPS-2-L paradoxically reduced richness vs. DSS (Figure 7A,B). DSS decreased Chao1 (richness) and Shannon (diversity) indices vs. the control (*p* < 0.05), with only BPS-2-H elevating the Shannon index (*p* < 0.05), aligning with the IBD-associated loss of diversity [2]. Beta diversity analysis (Figure 7C) revealed that BPS-2-M and BPS-2-H clustered near the controls, in contrast to BPS-2-L’s partial divergence. These dose-dependent effects parallel dendrobium polysaccharide-mediated colitis alleviation, confirming BPS-2-H’s efficacy in mitigating UC-linked dysbiosis [37].

Distinct gut microbiota profiles were observed across the groups. The dominant phylum in control mice were Firmicutes (37.5%) and Bacteroidota (53.1%), with Actinobacteria (2.5%), Desulfobacterota (3.6%), and Proteobacteria (2.5%) constituting 99.2% total abundance. DSS-induced colitis shifted dominance to Firmicutes (68.1%) and Bacteroidota (25%), with Proteobacteria (3.1%) totaling 96.2% (Figure 7D). BPS-2-H restored the phylum composition in UC mice to Firmicutes (35.9%), Bacteroidota (30.5%), and Verrucomicrobiota (30.4%), achieving 96.8% cumulative abundance. Dose-dependent reductions in Firmicutes were observed for the BPS-2-H group, which were close to control levels. Consistent with IBD clinical profiles [38], DSS-induced dysbiosis mirrored reduced microbial diversity with enriched pathogenic Proteobacteria, in contrast to the BPS-2-H restorative capacity. Notably, BPS-2-H significantly induced Verrucomicrobiota (30.4% vs. DSS 0.2%). This observation is clinically relevant due to its positive correlation with MUC2 mucin—a protective glycoprotein that attenuates gastrointestinal carcinogenesis [39]. Figure 7E demonstrates the DSS-elevated Firmicutes to Bacteroidota ratio (*p* < 0.05), recapitulating IBD-associated microbial signatures linked to inflammation [2]. BPS-2-H normalized this ratio (*p* < 0.05 vs. DSS group), demonstrating anti-inflammatory efficacy through modulation of the microbiota. The restored phylum-level similarity between BPS-2-H and control groups, particularly Verrucomicrobiota-mediated mucin enhancement, suggests BPS-2 polysaccharides mitigate colitis via microbial equilibrium restoration.

Genus-level analysis revealed significant alterations (Figure 7F). DSS mice showed reduced *Prevotellaceae*, *Alistipes*, *Enterobacter*, *Akkermansia*, and *Lactobacillus* (all *p* < 0.05 vs. control), with increased *Erysipelatoclostridium*, *Escherichia-Shigella*, *Bacteroides*, and *Eubacterium* (*p* < 0.05). High-dose BPS-2 reversed these shifts, elevating *Alistipes*, *Lactobacillus*, *Akkermansia*, *Turicibacter*, *Parabacteroides*, and *Parasutterella* while suppressing *Clostridium sensu stricto*, *Escherichia-Shigella*, *Eubacterium*, and *Bacteroides* (*p* < 0.05). Colon pathology scores inversely correlated with *Akkermansia* abundance. BPS-2-induced *Parabacteroides* enrichment promotes polysaccharide degradation and SCFA production, key anti-inflammatory mechanisms in IBD [2]. Elevated *Bacteroides* in DSS mice mirrors clinical IBD patterns, where its overabundance disrupts intestinal barriers and exacerbates inflammation [4].

LEfSe analysis identified differences in microbiota among groups (Figure 8A,B). Controls carried *Bacteroidales*, *Lactobacillus*, *Candidatus Saccharimonas*, *Alistipes*, and *Anaerofustis*. BPS-2-L enriched *Burkholderiales*, *Parasutterella*, and *Parabacteroides*; BPS-2-M increased *Lachnospiraceae* (family), *Erysipelotrichales*, *Eubacterium*, and *Turicibacter*; BPS-2-H elevated *Bacteroides*. *Bacteroidales*—dominant in healthy colonic microbiota—drive polysaccharide digestion and SCFA production via genera like *Parabacteroides* and *Bacteroides*, crucial for intestinal homeostasis [40]. BPS-2 enhanced *Bacteroidales* abundance, specifically by degrading the BPS-2 1,4-β-GalNAc/GlcNAc backbones into absorbable oligosaccharides. *Lachnospiraceae* was previously reported to exhibit robust polysaccharide degradation via β-galactosidase, α-L-arabinofuranosidase, α-amylase, and N-acetyl-glucosaminidase [41].

These findings suggest that BPS-2 polysaccharides ameliorate the intestinal environment in UC mice by modulating gut microbiota composition.

### 3.8. Model Construction and Molecular Docking of BPS-2 Oligosaccharide Fragment

The polysaccharide BPS-2 exhibits anti-digestive stability in gastrointestinal environments and modulates dysregulated gut microbiota in UC patients via defined microbial metabolism [16,17]. As it is challenging to isolate degradation products, a virtual screen was undertaken, which identified 26 bioactive oligosaccharide fragments (Appendix A) for molecular docking. As the link between polysaccharide anti-colitis effects and NF-κB pathway regulation is well established [13], interaction mechanisms between these oligosaccharides and key NF-κB signaling proteins were systematically investigated to elucidate structure–activity relationships.

Conformational, morphological, and structural characteristics revealed that the molecular backbone of BPS-2 is mainly composed of three monosaccharide units—namely glucosamine, galactosamine, and glucose—and exhibits a randomly coiled conformation (Appendix A). To further explore its biological mechanism, various parameters were quantified from the docking analysis, including binding free energy, the number of hydrogen bonds, and binding site information (Appendix A). Experimental data demonstrated that fragment BPS-2U exhibited significant binding advantages towards the NF-κB p50/p65 heterodimer, particularly with the p65 subunit: a binding free energy of −7.8 kcal/mol and the formation of 15 stable hydrogen bonds. The analysis of critical binding sites revealed that residues ARG50, SER51, LYS28, ARG30, GLU193, ASN186, GLU279, LYS195, ASP217, and LYS218 played pivotal roles in molecular recognition (Appendix A). Through energy minimization optimization and AutoDock-based docking simulations, fragment BPS-2U was successfully anchored within the bioactive pockets of both the NF-κB p50/p65 heterodimer and p65 monomer (Figure 9A,E). Notably, the strong binding of this oligosaccharide fragment to core components of the classical NF-κB signaling pathway suggests its potential regulatory function through interference with the dynamic equilibrium of the p50–p65–IκB trimer complex.

BPS-2U may inhibit NF-κB signal transduction via (1) competitive binding to the DNA-binding domain of the p65 subunit, thereby obstructing its post-nuclear translocation interaction with promoter regions of target genes; (2) stabilizing the binding conformation between IκBα inhibitory protein and the NF-κB complex, delaying IKK kinase-mediated phosphorylation and the subsequent ubiquitination-mediated degradation of IκBα; (3) directly interfering with the efficiency of the polyubiquitin chains assembly by the SCF-E3 ubiquitin ligase complex, thereby prolonging the half-life of IκBα. This multi-target mechanism mediates transcriptional suppression of pro-inflammatory cytokines, establishing a foundation to develop anti-inflammatory dietary supplements that mitigate inflammation [42].

### 3.9. Molecular Dynamics Simulation of the Oligosaccharide-p65 Protein Complex

The docking model of the p65–BPS-2U complex with the highest score was selected for molecular dynamics simulations (Figure 10). The root mean square deviation (RMSD), a critical indicator for evaluating the conformational stability of protein–ligand complexes, characterizes the deviation of atomic positions from the initial conformation, with lower values indicating superior conformational stability [32]. Simulation results (Figure 10A) revealed that the complex system reached dynamic equilibrium after a 30-ns equilibration period, with RMSD values stabilizing within a fluctuation range of 8 Å, confirming the stable binding characteristics of BPS-2U with the p65 target protein [43].

Further structural dynamics analysis demonstrated minor fluctuations in the radius of gyration (Rg) and solvent-accessible surface area (SASA) during the simulation (Figure 10B,C), suggesting conformational plasticity in the p65–BPS-2U complex during dynamic processes [44]. Notably, hydrogen bonding interactions played a pivotal role in maintaining ligand–protein binding stability. Dynamic trajectory analysis (Figure 10D) revealed that the number of hydrogen bonds between the complex fluctuated dynamically between zero and nine, with at least six hydrogen bonds maintained for approximately 70% of the simulation time. This indicated the formation of a stable hydrogen bond network between BPS-2U and the p65 protein. The root mean square fluctuation (RMSF) analysis of amino acid residue flexibility (Figure 10E) showed that the overall RMSF values of the complex system remained below 5 Å, particularly with lower fluctuation values observed in binding site residues, highlighting the high conformational rigidity and binding stability of the complex [32]. Integrated multidimensional analysis demonstrated that the p65–BPS-2U complex system not only exhibited exceptional dynamic stability but also achieved robust ligand–protein binding through multiple hydrogen bond interactions.

## 4. Conclusions

In this study, a DSS-induced murine UC model was developed to systematically evaluate the therapeutic effect of the polysaccharide BPS-2 on UC and to determine its underlying mechanism. BPS-2 significantly alleviated the pathological phenotypes of colitis. Histopathological analysis indicated that BPS-2 reduced colon damage by inhibiting the activities of MPO, COX-2, and iNOS in a dose-dependent manner. BPS-2 also induced the production of tight-junction proteins ZO-1, Occludin, and Claudin-1. In addition, BPS-2 coordinately induced antioxidant enzymes (SOD, CAT, and GSH-Px) and inhibited lipid peroxidation. BPS-2 improved the intestinal environment of colitis mice by modulating the composition of gut microbiota, in particular, through the enrichment of Bacteroidales and Lachnospiraceae. Molecular docking and dynamics simulation indicated that the BPS-2 oligosaccharide fragment BPS-2U preferentially bound to the NF-κB p50/p65 heterodimer.

This study systematically confirmed the remission effect of BPS-2 on UC at the in vivo level, establishing a preliminary foundation for its application. However, the existing data remains predominantly observational, and the deeper mechanisms—such as the regulation of gut microbiota and barrier restoration—are insufficiently elucidated. Future research will employ humanized colitis models to investigate the impact of BPS-2 on metabolic pathways within the gut microbiota. Concurrently, its active structure will be optimized via molecular stabilization (yielding BPS-2U) to enhance intestinal signaling regulation, combined with a microencapsulated delivery system for precise intestinal targeting. Multi-omics technologies and homeostasis models will be utilized to systematically validate its mechanism of action. Finally, building upon robust validation through comprehensive animal experiments, clinical collaborations will be pursued. This will provide a solid theoretical foundation and data support for the precise interventional application of BPS-2 in the food and pharmaceutical fields.

## Figures and Tables

**Figure 1 foods-14-02378-f001:**
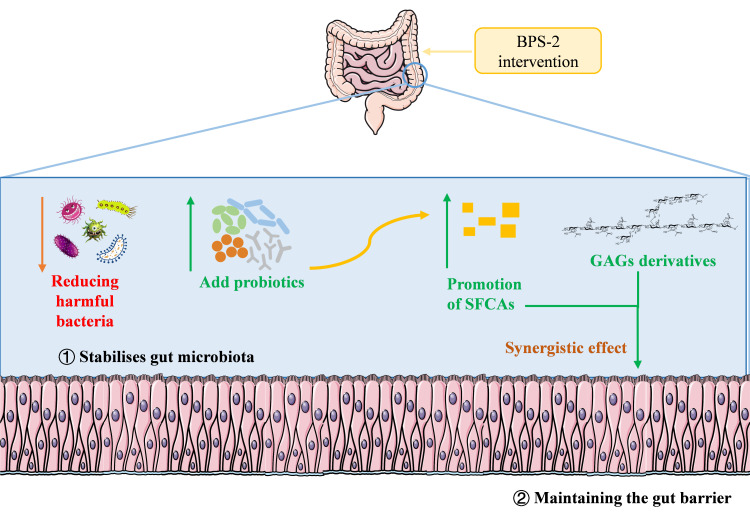
BPS-2 efficacy hypothesis schematic diagram.

**Figure 2 foods-14-02378-f002:**
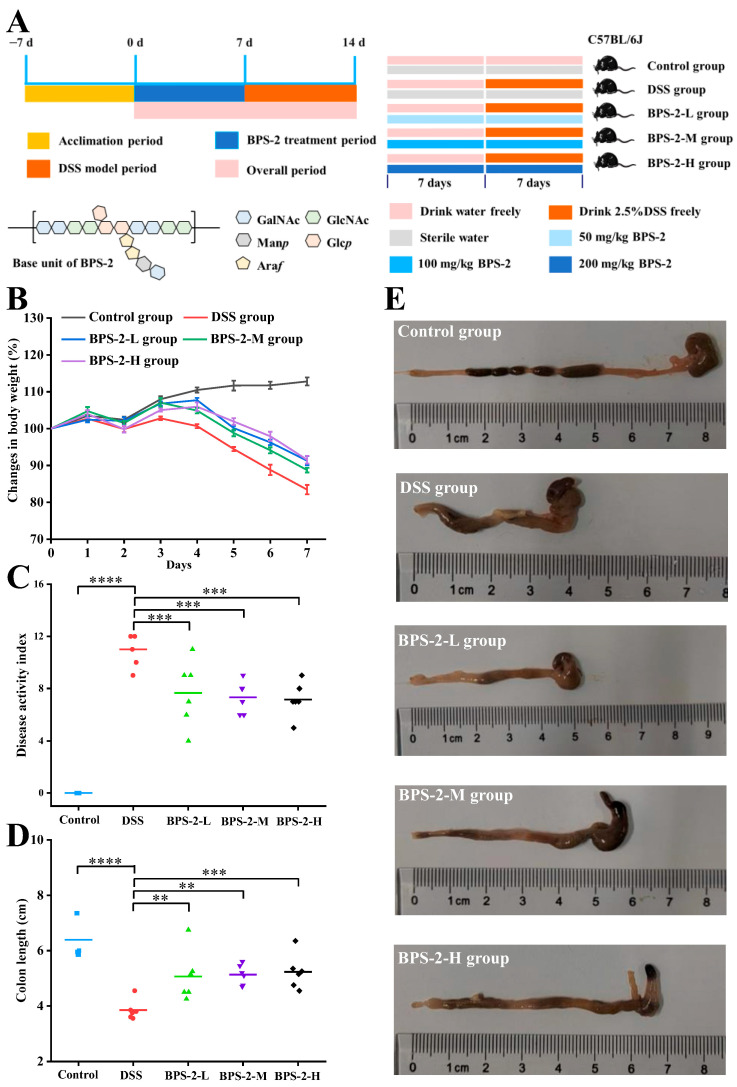
Establishment of a dextran sulfate sodium (DSS)-induced ulcerative colitis model in C57BL/6J mice and evaluation of BPS-2 intervention effects (*n* = 6). (**A**) Establishment of UC model in mice and BPS-2 administration; (**B**) Change in body weight (%); (**C**) Disease activity index; (**D**) Colon length; (**E**) Histological observations of mice colon. Statistical significance is defined as follows: **** *p* < 0.0001, *** *p* < 0.001, and ** *p* < 0.01.

**Figure 3 foods-14-02378-f003:**
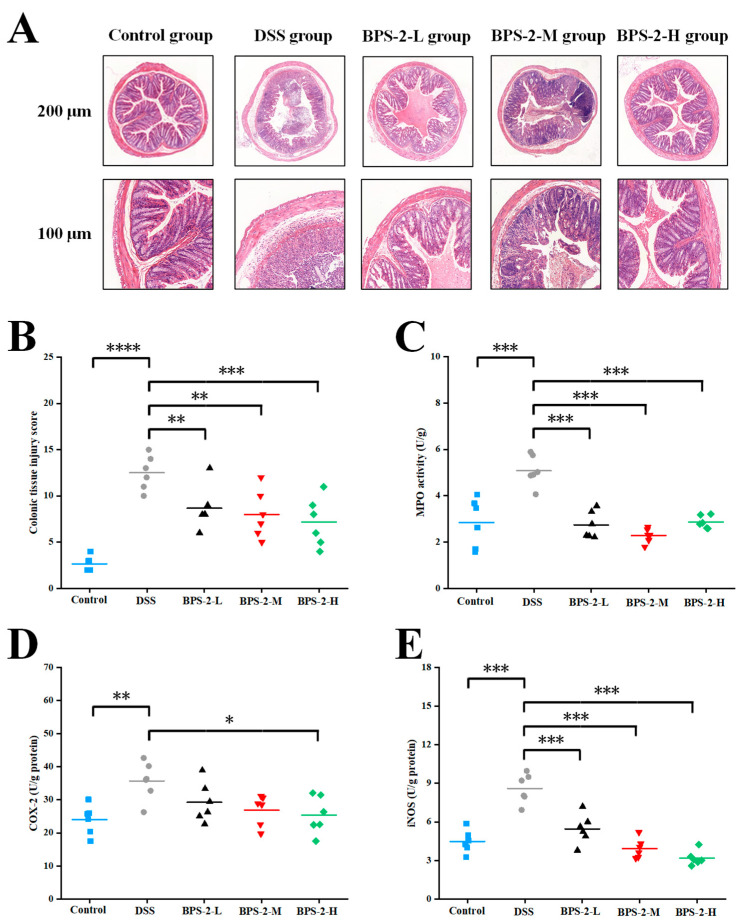
Effects of BPS-2 polysaccharide on colon tissue damage and enzyme activity in C57BL/6J mice with colitis (*n* = 6). (**A**) Microstructure observation; (**B**) Colon histological injury; (**C**) Myeloperoxidase (MPO); (**D**) Cyclooxygenase-2 (COX-2); (**E**) Inducible nitric oxide synthase (iNOS). Statistical significance is defined as follows: **** *p* < 0.0001, *** *p* < 0.001, ** *p* < 0.01, and * *p* < 0.05.

**Figure 4 foods-14-02378-f004:**
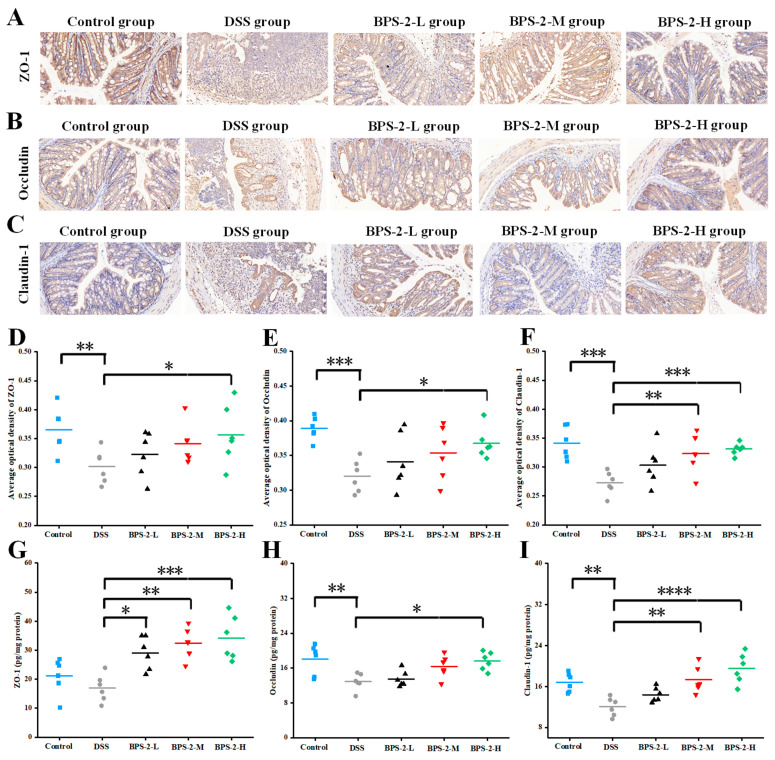
Effects of BPS-2 polysaccharide on tight junction (TJ) proteins in C57BL/6J mice with colitis (*n* = 6). Immunohistochemical staining of (**A**) Zonula occludens-1 (ZO-1); (**B**) Occludin; (**C**) Claudin-3. Bar graphs show the mean optical density of (**D**) ZO-1; (**E**) Occludin; (**F**) Claudin-3 in the colon. (**G**) ZO-1; (**H**) Occludin; (**I**) Claudin-3 expression in colon tissue. The scale bar in the immunohistochemical image represents 50 μm. Statistical significance is defined as follows: **** *p* < 0.0001, *** *p* < 0.001, ** *p* < 0.01, and * *p* < 0.05.

**Figure 5 foods-14-02378-f005:**
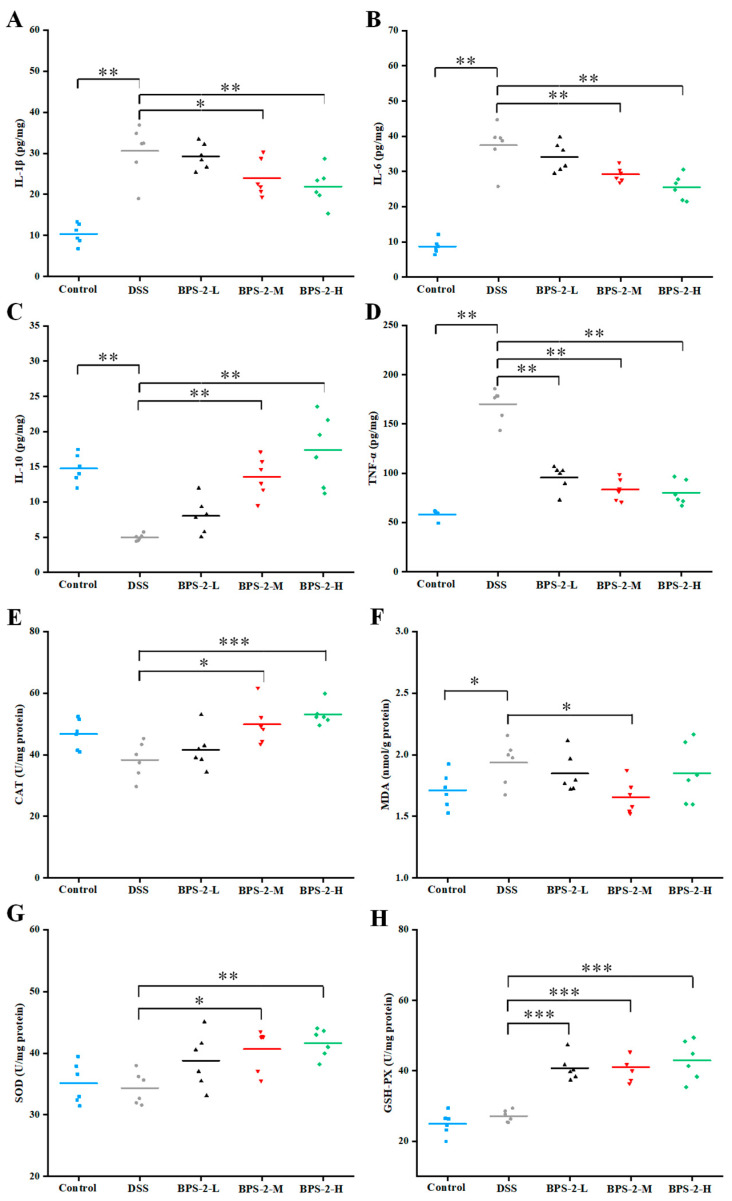
Effects of BPS-2 polysaccharide on cytokines and enzyme activity in colon tissue of C57BL/6J mice with colitis (*n* = 6). (**A**) Interleukin-1 beta (IL-1β); (**B**) Interleukin-6 (IL-6); (**C**) Interleukin-10 (IL-10); (**D**) Tumor necrosis factor alpha (TNF-α); (**E**) Catalase (CAT); (**F**) Malondialdehyde (MDA); (**G**) Superoxide dismutase (SOD); (**H**) Glutathione peroxidase (GSH-Px). Statistical significance is defined as follows: *** *p* < 0.001, ** *p* < 0.01, and * *p* < 0.05.

**Figure 6 foods-14-02378-f006:**
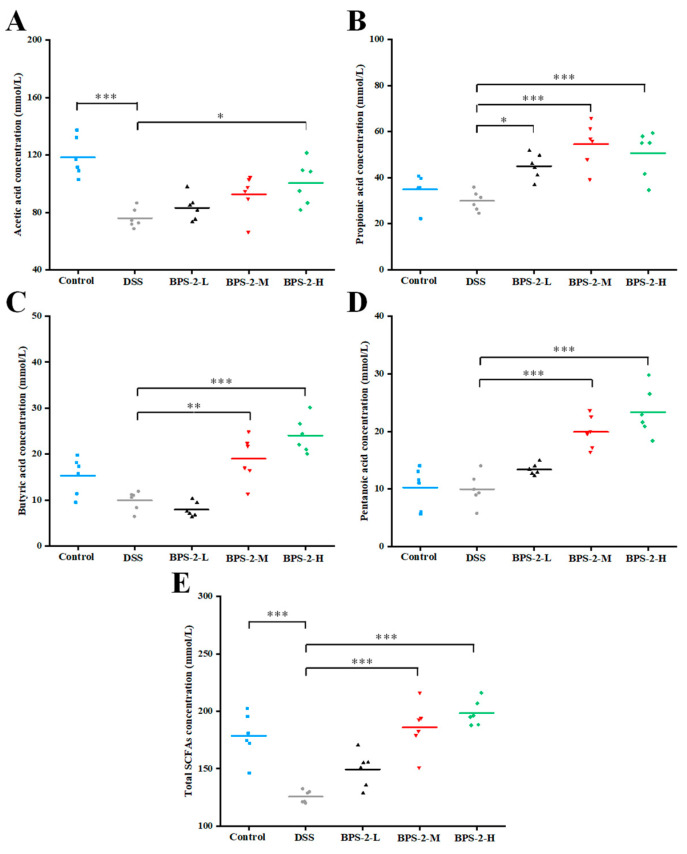
Effects of BPS-2 polysaccharide on the production of short-chain fatty acids (SCFAs) in C57BL/6J mice with colitis (*n* = 6). (**A**) Acetic acid; (**B**) Propionic acid; (**C**) Butyric acid; (**D**) Pentanoic acid; (**E**) Total SCFAs. Statistical significance is defined as follows: *** *p* < 0.001, ** *p* < 0.01, and * *p* < 0.05.

**Figure 7 foods-14-02378-f007:**
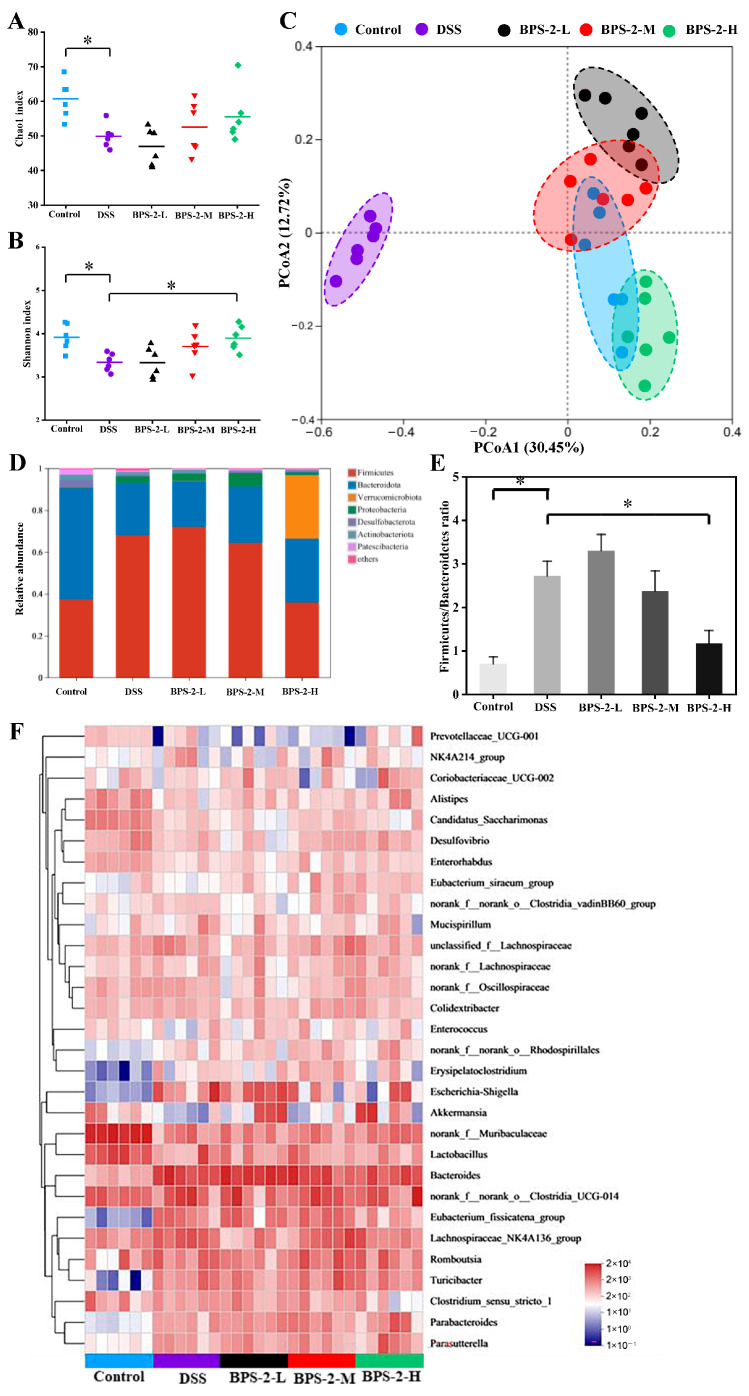
Effects of BPS-2 polysaccharide on the diversity of gut microbiota and colonic microorganisms in C57BL/6J mice with colitis (*n* = 6). Significant differences were observed between the dextran sulfate sodium (DSS) group and each dose group (*n = 6*). (**A**) Chao1 index; (**B**) Shannon index; (**C**) β-diversity of gut microbiota; (**D**) Changes in relative abundance of gut microbiota at the phylum level; (**E**) Changes in Firmicutes/Bacteroidetes ratio; (**F**) Heat map of bacterial relative abundance at the genus level (top 30 of species). Statistical significance is defined as * *p* < 0.05.

**Figure 8 foods-14-02378-f008:**
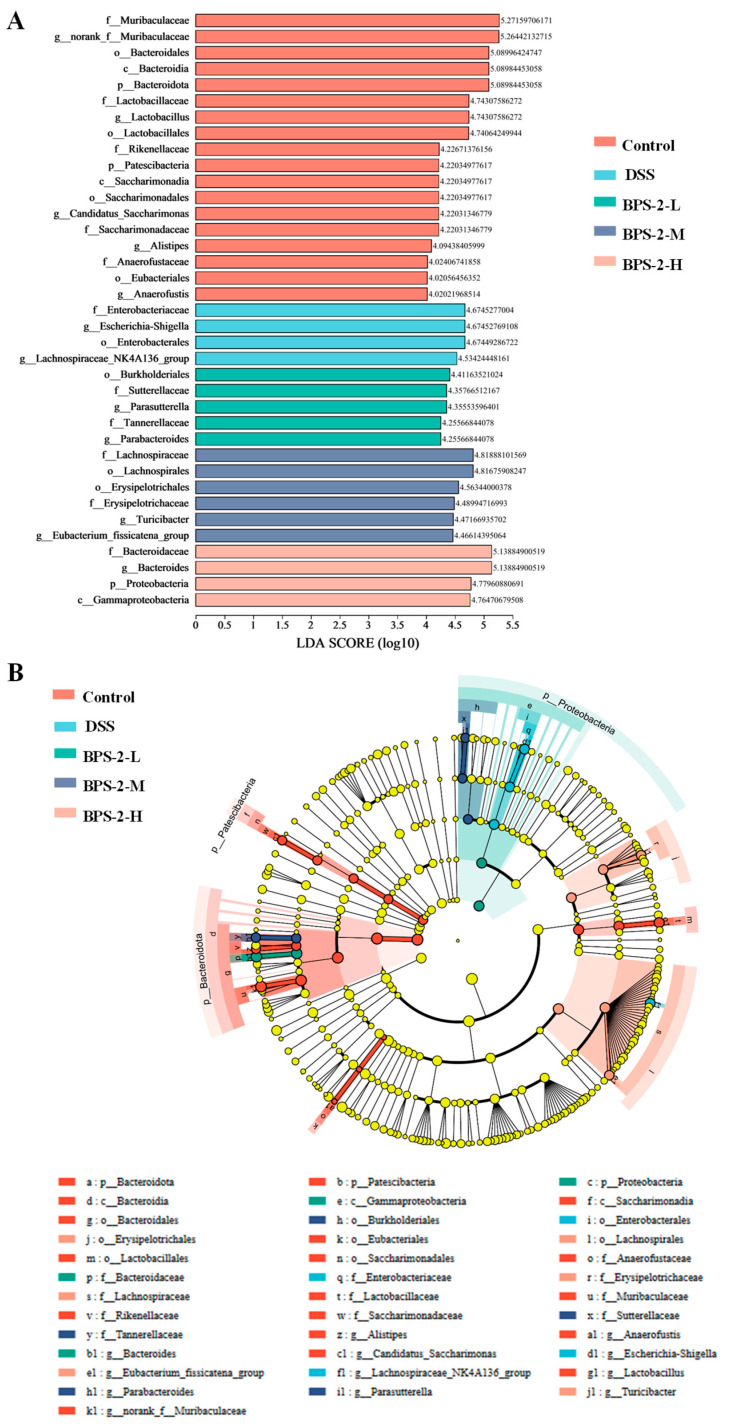
Linear discriminant analysis Effect Size (LEfSe) was used to identify differentially abundant gut microbiota taxa (Linear discriminant analysis (LDA) score threshold > 4.0). (**A**) LDA score distribution histogram; (**B**) LDA score distribution cladogram.

**Figure 9 foods-14-02378-f009:**
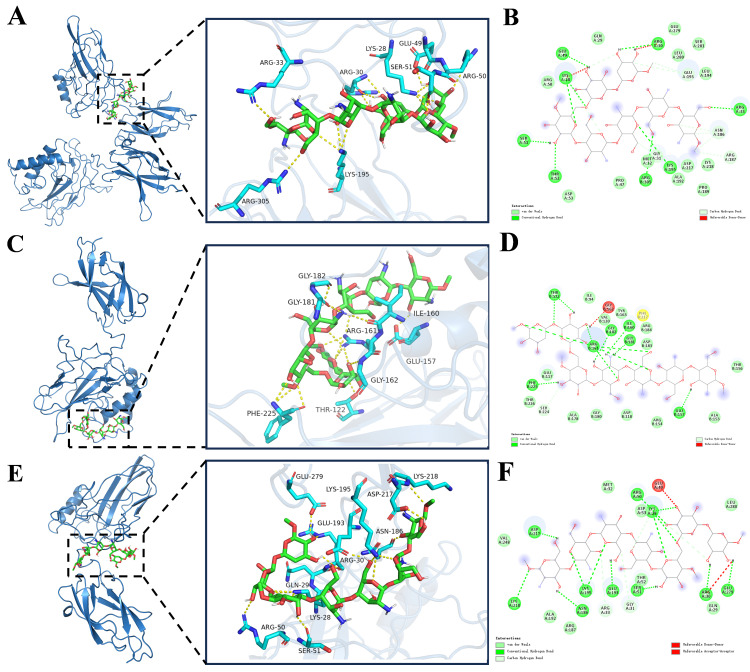
Molecular docking of oligosaccharide fragment chain with Nuclear Factor κB (NF-κB), p50, and p65 protein molecules. (**A**,**C**,**E**) are the highest-scoring three-dimensional docking model positions of NF-κB, p50, and p65 protein molecules; (**B**,**D**,**F**) represent two-dimensional interaction diagrams of the docking models for NF-κB, p50, and p65 protein molecules.

**Figure 10 foods-14-02378-f010:**
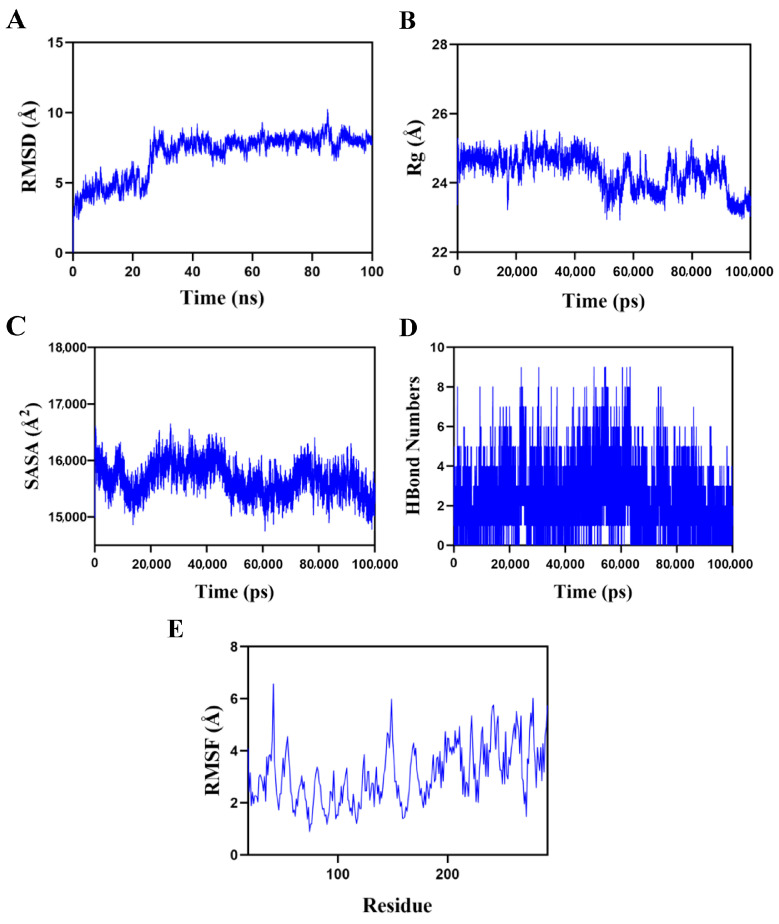
Molecular dynamics simulation of the protein–ligand complex. (**A**) Root mean square deviation (RMSD) values of the p65–BPS-2U complex over time; (**B**) Radius of gyration (Rg) values of the p65–BPS-2U complex over time; (**C**) Solvent-accessible surface area (SASA) values of the p65–BPS-2U complex over time; (**D**) H-Bond values of the p65–BPS-2U complex over time; (**E**) Root mean square fluctuation (RMSF) values of amino acid backbone atoms in the p65–BPS-2U complex over time.

## Data Availability

The original contributions presented in the study are included in the article/Appendix A, further inquiries can be directed to the corresponding authors.

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
