# Peer review of "Bacillus thuringiensis Exopolysaccharide BPS-2 Ameliorates Ulcerative Colitis in a Murine Model Through Modulation of Gut Microbiota and Suppression of the NF-κB Cascade"

_foods, 2025, doi:10.3390/foods14132378_

Round 1
Reviewer 1 Report
Comments and Suggestions for Authors
The authors investigated the therapeutic potential of the exopolysaccharide BPS-2 derived from Bacillus thuringiensis in a dextran sulfate sodium (DSS)-induced ulcerative colitis (UC) murine model. They demonstrated that BPS-2 alleviated UC symptoms by preserving colon epithelial integrity, restoring gut microbiota balance, and suppressing inflammatory responses via inhibition of the NF-κB signaling cascade. BPS-2 treatment enhanced tight junction protein expression, reduced oxidative stress and promoted short-chain fatty acid production. Molecular docking and dynamics simulations revealed that the BPS-2 oligosaccharide fragment BPS-2U binds stably to the NF-κB p65 subunit, indicating a direct mechanism for its anti-inflammatory effects. Overall, the study highlights BPS-2 as a promising food-derived agent for gut-targeted UC therapy.
-It is unnecessary to reintroduce abbreviations (e.g UC, DSS) as they have already been defined in the abstract. Additionally, abbreviations should not be used before they are first introduced in the text (e.g., NF-κB). Please revise accordingly.
-The introduction section would benefit from the inclusion of a simple schematic summarizing the proposed or investigated mechanisms by which BPS-2 influences inflammation, oxidative stress, tight junction integrity, and gut microbiota. This would help enhance reader comprehension and provide a clearer conceptual framework for the study.
-In lines 222 and 224, please provide the full species names for B. circulans and B. subtilis.
-The text within Figure 1, particularly the parts A and B, is too small and difficult to read. Please update it to ensure a consistent font size and format for better readability. The same issue also applies to F, G and H parts of the Figure 6.
-All the figures and tables in a manuscript should be cited within the text before their first appearance. Please revise the placement of citations for Figures 1 and 5 accordingly.
-In line 306, the text "vs" appears in yellow. Please check it.
-Line 496, please insert a space at the end of the sentence to maintain formatting consistency.
-In line 556, the “RMSF” is used before its full form is introduced. Please revise to ensure the full term appears first.
Author Response
For the detailed response, please refer to the attachment.

Reviewer 2 Report
Comments and Suggestions for Authors
The Manuscript entitled “Bacillus thuringiensis exopolysaccharide BPS-2 ameliorates ulcerative colitis in a murine model through modulation of gut microbiota and suppression of the NF-κB cascade” describes the therapeutic potential of bacterial extracellular polysaccharide, BPS-2 from Bacillus thuringiensis in murine mode of ulcerative colitis. The authors were able to provide scientific evidence using mouse model that BPS-2 protect the colon by attenuating histological damage, modulating immune marker expression pattern, ultimately restoring gut microbiota homeostasis. The authors were successful in collecting supportive data to show the nutraceutical effects of BPS-2, which opens a novel avenue in the development of novel functional prebiotic ingredient to alleviate ulcerative colitis and enhance gut health. Hence, the work is novel, well executed and conclusions are supported by the data. Overall, the manuscript clearly delineates the purpose of research and is acceptable for publication with minor corrections.
There are some points that need to be addressed by the authors to further improve the quality of the manuscript.
- Page no. 3 line no.125: …daily via intragastric administration on…Does the authors want to tell the reader that BPS-2 is administered to the mouse using a gavage method or some other method? It seems to be confusing, in line no. 122 authors mentioned that saline is administered using gavage method. Please clarify this and try to use uniform words throughout the manuscript for more clarity.
- Page no. 4 line no. 150 “HE stained sections” what is the meaning of this, if authors are referring this to Haematoxylin and eosin (H&E) staining; if an abbreviation is used in the manuscript, the first time it should be written as the expanded version along with the abbreviation in the parentheses. Subsequently, for each time it is repeated within the manuscript, the abbreviation should be sufficient throughout the manuscript. Authors are requested to follow this rule for each abbreviation used in the manuscript.
- The authors arranged all the figures satisfactorily. However, figure legends need some modifications. Figure legends must be standalone with the corresponding figure. Also, figure legends should always be accompanied by the expanded version of each abbreviation that is used in that specific figure.
- In figures, the bar graphs should show the individual mouse point/value within the solid bars along with the error bars wherever it is possible, especially Figure 3D. This may help the reader to understand the variability among animals within the experimental groups
- In Figure 4 authors show that enzyme activities of Super oxide dismutase (SOD), Glutathione peroxidase (GSH-PX) and Catalase (CAT) are restored by BPS-2 treatment. It seems that all three enzyme activities are improved in such a way that it is higher than that of the control mice. Authors are requested to explain this. How can these enzyme activities go way higher than the normal homeostasis? Also rewrite this figure legend, it’s not enzyme expression instead it is the enzyme activity.
- It would be better if they could add a section titled “strengths and limitations of the study” and list out all the strengths and limitations of this study in this section.
Author Response

(The authors gave the same response as above.)

Round 2
Reviewer 1 Report
Comments and Suggestions for Authors
All requested changes have been fully addressed in the revised manuscript.
Author Response
We sincerely appreciate your positive feedback on our revised manuscript. Wishing you all the best in your future endeavors.